# Finding Children with High Risk of Non-Vaccination in 92 Low- and Middle-Income Countries: A Decision Tree Approach

**DOI:** 10.3390/vaccines9060646

**Published:** 2021-06-13

**Authors:** Thiago M. Santos, Bianca O. Cata-Preta, Cesar G. Victora, Aluisio J. D. Barros

**Affiliations:** International Center for Equity in Health, Federal University of Pelotas, Marechal Deodoro, 1160, Pelotas 96020-220, Brazil; bcatapreta@equidade.org (B.O.C.-P.); cvictora@equidade.org (C.G.V.); abarros@equidade.org (A.J.D.B.)

**Keywords:** vaccination, immunization, prenatal care, decision trees

## Abstract

Reducing vaccination inequalities is a key goal of the Immunization Agenda 2030. Our main objective was to identify high-risk groups of children who received no vaccines (zero-dose children). A decision tree approach was used for 92 low- and middle-income countries using data from Demographic and Health Surveys and Multiple Indicator Cluster Surveys, allowing the identification of groups of children aged 12–23 months at high risk of being zero dose (no doses of the four basic vaccines—BCG, polio, DPT and measles). Three high-risk groups were identified in the analysis combining all countries. The group with the highest zero-dose prevalence (42%) included 4% of all children, but almost one in every four zero-dose children in the sample. It included children whose mothers did not receive the tetanus vaccine during and before the pregnancy, who had no antenatal care visits and who did not deliver in a health facility. Separate analyses by country presented similar results. Children who have been missed by vaccination services were also left out by other primary health care interventions, especially those related to antenatal and delivery care. There is an opportunity for better integration among services in order to achieve high and equitable immunization coverage.

## 1. Introduction

With the end of the decade, the Global Vaccine Action Plan 2011–2020 oversaw important advances in vaccination coverage, especially in low- and middle-income countries (LMICs), but many of its targets have not been met [1]. Between 2010 and 2018, it was estimated that the measles vaccine prevented about 23 million deaths, but large measles outbreaks occurred in all World Health Organization (WHO) regions in 2018 [1,2]. Since 2010, 116 countries have introduced vaccines not previously used, but coverage of essential vaccines have stagnated and polio has not yet been eradicated worldwide [1,3]. The COVID-19 pandemic, regional outbreaks of diseases such as the Ebola Virus Disease, and the threat of new pandemics represent an important challenge for health systems on the continuation and improvement of routine vaccination services in all countries [3]. 

The Immunization Agenda 2030 (IA2030) recognized the contribution of vaccination in 14 of the 17 Sustainable Development Goals (SDGs), especially SDG 3: “Ensure healthy lives and promote well-being for all at all ages.” With over 13 million children receiving no vaccines each year, the IA2030 selected “leave no one behind” as a motto and adopted a strategy that should target both reduction in inequalities and be tailored to the national context of each country [3]. Identifying risk groups for non-vaccination in LMICs could be useful for reducing inequalities and tailoring interventions that could reach and successfully vaccinate the children that are being left behind. Zero-dose children—those who failed to receive any vaccine—represent a particularly vulnerable group being left behind by multiple vaccination services and interventions.

Vaccination estimates that use administrative data tend to be less reliable, due to inaccuracies in the population denominator and the reported number of doses, and survey data are often considered a more reliable alternative [4]. Demographic and Health Surveys (DHS) and Multiple Indicator Cluster Surveys (MICS) are nationally representative household surveys that collect information on both vaccination and sociodemographic indicators that can be used for identifying risk groups of zero-dose children and their characteristics. According to 241 DHS and MICS surveys from 1986 to 2007 from 96 LMICs, the percentage of zero-dose children was 9.9% among those aged 12–59 months. Zero-dose status was generally more common in children from the poorest households and from mothers who were less educated, not vaccinated with the tetanus toxoid vaccine and who decided alone about child’s care when the child was sick [5].

In order to identify high-risk groups of zero-dose children, the analysis of sample subgroups (according to indicators of wealth or education, for example) is a common option, but it tends to be restricted in terms of the number of subgroups tested and the different combinations of indicators that are used for stratification. As the number of combinations and indicators grow, the analysis gets progressively more complex and the sample size smaller, hindering the identification of high-risk groups. A possible alternative is the use of decision trees. They are robust statistical tools that provide a framework for subgroup identification, allowing for a greater number of indicators and their combinations to be tested, and without the need to make estimates for all possible combinations [6]. In the context of non-vaccination, decision trees can be used for identifying subgroups of children at a higher risk of being zero dose, without previous specification of what subgroups should be tested. 

Our objective is to identify high-risk groups of zero-dose children in LMICs, using a decision tree approach, in order to determine defining features that can be used for both targeting and tailoring vaccination policies and interventions at national level and globally.

## 2. Materials and Methods

### 2.1. Data Source and Children Sample

We analyzed 210,509 children from 92 LMICs with DHS or MICS surveys conducted since 2010 and with data released until July 2020. For each country, we selected the most recent survey with available information for the Bacillus Calmette-Guérin (BCG), poliomyelitis (polio), diphtheria, tetanus, pertussis (DTP), and measles vaccines. DHS and MICS are household surveys whose responsible agencies—respectively the United States Agency for International Development (USAID) and the United Nations Children’s Fund (UNICEF)—collaborate to ensure that both are harmonized and comparable [7]. We classified each country according to their World Bank income level using the median survey year, 2015 [8].

The proportion of LMICs represented by the surveys in our study in each UNICEF region was 92% in West and Central Africa, 79% of Eastern and Southern Africa, 58% of Middle East and North Africa, 55% of Eastern Europe and Central Asia, 75% of South Asia, 48% of East Asia and the Pacific, and 58% of Latin America and Caribbean. Regarding the World Bank income level in 2015, the proportions were 90%, 73%, and 46% of the low-, lower-middle, and upper-middle income countries, respectively. 

### 2.2. Outcome

We assessed the vaccination status for each child using information from vaccination cards and, when the card was not available or was not shown, using caregiver reports. We analyzed children aged 12–23 months living with their mothers [9]. Exceptions were Moldova (15–26 months) and Bosnia and Herzegovina, Costa Rica, Jamaica, North Macedonia and Ukraine (18–29 months), due to the fact that the measles vaccine was given at 15 and 18 months in those countries during the survey year, respectively. 

The studied outcome was the zero-dose status of the child, defined as not having received any doses of the BCG, polio, DPT, and measles vaccines. In case of missing information for a vaccine, the child was considered as not having received it, according to the WHO recommendation [9]. 

### 2.3. Potential Indicators of Vulnerability

We selected 15 potential indicators that were available in both DHS and MICS surveys, that have been previously used for zero-dose investigation in the literature [5,10,11] and that can provide useful information for targeting and tailoring vaccination interventions. 

The household indicators from the literature were: number of household members, sex of household head, place of residence (urban or rural), wealth quintiles (from poorest to richest: Q1, Q2, Q3, Q4, and Q5), religious affiliation (classified as Christian, Muslim, Hindu, Buddhist, Folk, other, or unaffiliated), and ownership of radio or television. The maternal indicators were: education (classified as none, primary or secondary+), marital status (classified as never married or in a union, currently married or in a union, or formerly married or in a union), and maternal age in complete years. We also included the number of live born children that had died to date. The child and pregnancy indicators were: sex of the child, number of antenatal care visits during the pregnancy, place of delivery (classified as noninstitutional, institutional–public, or institutional–private), and the number of tetanus toxoid injections received before and during the pregnancy.

Additionally, we included two other household indicators: ownership of a refrigerator and the number of living children, resulting in 17 indicators. Ownership of a refrigerator was included as a form of negative control to radio and television, since the three are indicators of wealth, but, different from the other two, a refrigerator is not a source of information. The number of living children was included as a possible indicator of competition for time and household resources necessary for vaccination.

The wealth quintiles are based on a wealth index created by the MICS and DHS teams via principal components analysis (PCA) of household indicators: building materials, ownership of appliances, presence of electricity, water and sanitation, among others [12,13]. Separate PCAs are carried out for urban and rural settings in order to take into account the differences in relevance that household assets can have in those settings. The households are then divided into quintiles according to their wealth index [14].

### 2.4. Analysis Methods

We calculated the zero-dose prevalence and its 95% confidence interval (95% CI) for each country taking into account the complex survey design used by DHS and MICS using Stata (StataCorp. 2019. Stata Statistical Software: Release 16. College Station, TX: StataCorp LLC) [7] and created a world map to present those prevalences using R (R Core Team, 2020, version 4.0.2. R Foundation for Statistical Computing, Vienna, Austria) and publicly available geographics datasets [15]. 

In order to identify the groups of children with the highest zero-dose prevalence in the studied countries combined and for each individual country, we used a decision tree approach.

The most frequently used technique for creating decision trees is the Classification and Regression Tree (CART) method [6]. For the identification of groups with high risk, CART performs a binary recursive partitioning process. This means that, using the indicators available for the analysis, it divides the sample into two groups (hence binary), one with a higher zero-dose prevalence and one with a lower, when compared to the original complete sample. The algorithm tests all the indicators and possible cut-off points (e.g., belonging to Q1 vs. Q2–Q5, belonging to Q1–Q2 vs. Q3–Q5, and so on). The best split is the one that maximizes within-group homogeneity. This process eliminates the necessity of previously specifying cut-off points for non-binary indicators and also allows for a specific stratification to happen within different subgroups of the sample, dealing naturally with the complex interactions of indicators.

This process will go on, splitting the groups in the previous iteration into smaller groups (hence recursive) until a stopping rule halts the process. The stopping rule can be, for example, a minimum number of children in one subgroup, also called a node. 

Since the outcome is a binary indicator, CART creates a classification tree, meaning a tree that classifies each child as “zero dose” or “nonzero dose” according to the terminal node the child is assigned. However, the zero-dose prevalence in most surveys is much smaller than 50%, creating an unbalanced sample for the classification algorithm. This can be thought of as using a diagnostic tool to identify people with a rare disease. The tree can simply classify all children as “nonzero dose” and still have high accuracy, 100% specificity, but 0% sensitivity. In order to mitigate this imbalance, we adjusted the misclassification cost used by CART. During the creation of the tree, classifying a “zero dose” child as “nonzero dose” (false negative) costed double than a false positive, reducing the threshold necessary for the children in a node to be classified as zero dose. A more detailed description of how misclassification costs are used in CART can be found elsewhere [16,17]. We made this specification in order to increase the tree sensitivity, even if accompanied by a reduction in specificity. Moreover, any split leading to a node with less than 50 children was discarded.

We created a decision tree for each of the 92 countries separately and one for all countries combined using the CART implementation of the rpart package version 4.1-15 [17] in R (R Core Team, 2020, version 4.0.2. R Foundation for Statistical Computing, Vienna, Austria). All the trees were created using sample weights that consider the complex survey design. The individual trees are presented in country profiles that also include the prevalence of key inequality and healthcare indicators. For the pooled analyses (prevalence and decision trees), we combined the datasets and adjusted the sample weights in order to take into consideration the number of children aged 12–23 months living in each country, as described in Appendix A. 

We analyzed the composition of each risk groups identified by the pooled tree in terms of wealth, maternal education, place of residence, and sex of the child in order to better characterize inequalities among these groups. This is an important complementary equity analysis to the decision tree since the indicators featured in the tree are the ones best suited for identifying the zero-dose groups, but this does not mean that there are no differences in terms of other indicators. 

## 3. Results

### 3.1. Zero-Dose Prevalence

The pooled zero-dose prevalence was 7.7% (95% CI 7.4–7.9%), but varied widely across all countries, with a median prevalence of 2.9% (interquartile range: 1.5–8.7%). Moldova had the lowest prevalence, with no zero-dose children identified in the sample. Meanwhile, South Sudan had the highest prevalence: 57.6% (95% CI 54.1–61.0%); followed by Kiribati: 38.2% (95% CI 33.4–43.4%); Papua New Guinea: 24.2% (95% CI 20.9–27.9%); Guinea: 23.0% (95% CI 19.8–26.4%); and Congo, DR: 20.5% (95% CI 17.9–23.3%). The national prevalence level for all countries is presented in Figure 1 and confidence intervals in Appendix A in Appendix A.

### 3.2. Pooled Decision Tree

The decision tree created for all countries combined divided the sample into four groups of children with vastly different zero-dose prevalence (Figure 2). Of all the 17 indicators included in the tree analyses, only three were selected to separate the children into risk groups: the number of tetanus vaccine doses received by the mother during or before the pregnancy of the child, the number of antenatal care visits, and the child’s place of delivery. 

For each group, we presented three proportions: (1) Zero-dose prevalence, i.e., the proportion of children in that specific group who were zero dose; (2) % of all children, i.e., the proportion of all children in the sample assigned to that group; and (3) % of all zero-dose children, i.e., the proportion of all zero-dose children in the sample assigned to that group.

Group 1 was composed of children whose mother received at least one dose of the tetanus vaccine. It includes 90% of all children and also the majority of all zero-dose children (67%), given it is a very large group. But it was the group with the lowest zero-dose prevalence: 5.7% (95% CI 5.5–5.9%).

Group 2 was composed of children whose mothers received no tetanus vaccine, but who had at least one antenatal care visit during the child’s pregnancy. It includes 5% of all children and 7% of all zero-dose children. It had almost double the zero-dose prevalence of Group 1: 11.0% (95% CI 9.9–12.3%).

Group 3 was composed of children whose mothers received no tetanus vaccine and had no antenatal care visits, but who were born in a health facility. It includes 1% of all children and 2% of all zero-dose children. It had the second highest zero-dose prevalence among the groups: 22.6% (95% CI 18.6–27.2%).

Group 4 was composed of children whose mothers received no tetanus vaccine and had no antenatal care visits and whose delivery was not in a health facility. With only 4% of all children in this group, it includes 24% of all zero-dose children and a zero-dose prevalence of 42.2% (95% CI 39.7–44.8%).

### 3.3. Characterization of the Risk Groups

There was a clear pattern of inequality between the four groups: the higher the zero-dose prevalence in a group, the poorer, more rural, and with a less educated mother the children in that group. In Groups 1 to 4, respectively, 21.7% (95% CI 21.3–22.1%), 22.0% (95% CI 20.0–24.1%), 29.5% (95% CI 25.2–34.2%), and 47.0% (95% CI 44.1–50.0%) of the children belonged to the poorest wealth quintile; 63.3% (95% CI 62.7–63.8%), 61.0% (95% CI 58.8–63.2%), 70.1% (95% CI 64.2–75.4%), and 89.3% (95% CI 87.7–90.8%) lived in a rural area; and 22.1% (95% CI 21.7–22.5%), 32.0% (95% CI 29.6–34.5%), 52.2 (95% CI 47.1–57.2%), and 80.9% (95% CI 79.0–82.6%) had a mother with no education. A comparison between the four groups and all children combined is presented in Figure 3. There were no statistically significant differences in terms of sex of the child among any groups.

### 3.4. Country-Specific Trees

Of the 92 countries studied, it was possible to identify high-risk groups in only 25 with the decision tree approach. The median zero-dose prevalence for the 67 countries for which no high-risk groups were identified was 1.7% (ranging from 0.0% to 16.3%) and the median sample size was 1337 children (ranging from 266 to 5820). For the other 25 countries, the median prevalence was 13.0% (ranging from 3.3% to 57.6%) and the median sample size was 2151 children (ranging from 453 to 49,284). 

Among the countries with risk group separation, 15 featured the number of antenatal care visits in their tree, 10 the place of delivery, 8 the number of tetanus vaccine doses, and also 8 the wealth quintile. The results for all indicators are presented in Table 1. The 92 trees are presented in the country profiles in Appendix A in Appendix A.

## 4. Discussion

The pooled decision tree indicates a clear message: the children at a higher risk of being zero dose are the ones who themselves and whose mothers have been left out by other health services and interventions, in particular antenatal care. The fewer health services and interventions the child/mother pair received, the higher the risk that the child has not been vaccinated. The triple exposed children (those whose mother did not receive any doses of the tetanus vaccine, who did not have any antenatal care visits and whose delivery was not in a health facility) had an alarming zero-dose prevalence of 42%. But it is important to stress: no causal relationship is established by the decision tree. The analysis only indicates that both phenomena have happened in the same household.

The fact that the triple exposed children were only a small percentage of the sample (4%), but they accounted for one in every four zero-dose children indicates: (1) the opportunity they represent for targeted interventions and (2) how challenging it is to reach them, as they and their mothers have already been left out by vaccination, antenatal care, and delivery services. This result corroborates the current recommendation by the IA2030 to “Encourage greater collaboration and integration within and beyond the health sector,” reinforcing the importance of integration between primary health care services in order to increase efficiency and reach those who are being left out by multiple basic interventions [3].

It is no surprise that the first indicator selected by the tree was the mother not having received any doses of the tetanus vaccine, which could evidence similar challenges in effectively vaccinating mother and child, involving vaccine supply, logistics, application, monitoring, and long-term predictable funding [18]. Both vaccination and antenatal care services failed to reach and vaccinate these mothers during and before the index pregnancy, which contributes to the burden of maternal and neonatal tetanus [19]. Furthermore, one in every four children of these mothers was also zero dose, therefore failing to achieve both short and long-term protection against tetanus for the child. 

The lack of any antenatal care visits can also function as an indicator of families that are harder to reach by health services, but it can also have an impact on the future vaccination of the child. Mothers with more antenatal care visits have more opportunities to receive positive messages by healthcare providers regarding the advantages of vaccinating their soon-to-be-born child, increasing awareness of the benefits and safety of vaccines and sharing information such as the appropriate place and time for vaccination [20]. 

The child not being delivered in a health facility can have a more direct link with the zero-dose status. The WHO recommends that in countries with high burden of tuberculosis a single dose of the BCG vaccine should be given at birth to all healthy neonates [21] and 152 LMICs have a policy of universal neonatal vaccination at birth or at the first week of life [22]. This is in accordance with our results: among children whose mothers have not been vaccinated against tetanus and who have had zero antenatal care visits, those who were born in a health facility had a zero-dose prevalence of 23% while those who were not had almost double the prevalence (42%).

Our equity analyses showed that the groups of children at higher risk of being zero dose were significant and incrementally poorer, more rural and with less educated mothers. This is in line with the literature, as children in those conditions are less likely to be vaccinated [5] and their mothers are also less likely to receive qualified antenatal care [23]. This result further indicates the level of vulnerability of the children in the higher risk groups, as their socioeconomic condition has also been associated with lower medical-treatment-seeking behavior [24], worse nutritional status [25], and higher mortality among unvaccinated children [26]. 

In 2002, the WHO and the UNICEF launched the “Reach Every District” (RED) strategy, an initiative that promotes the prioritization of districts with poor access and utilization of vaccination services [27], with reports of increased vaccination coverage [28] and detection of vulnerable populations [29,30] after the adoption of the RED strategy. A proposed improvement to the RED strategy is the “reach every community” approach, with a focus on facility-level planning, monitoring of community access and utilization of vaccination services, local communication strategies and health networks that should allow for the tailoring of vaccination delivery systems according to the specific necessities of vulnerable communities [29]. Those strategies are an opportunity to promote other maternal, newborn, and child health (MNCH) services, since community involvement can help to identify newborn and pregnant women that could be targets for those services [31]. An example is the inclusion of a wider MNCH, environmental health, and water and sanitation package by planners involved in the application of the RED strategy in the Byanzurkh District in Mongolia, since they determined that both vaccination and MNCH services shared similar barriers in the district (distance, level of education, and poverty) [30].

The strengths of this paper include: the multi-country approach, with the inclusion of 92 nationally representative surveys and the resulting large sample size; the choice of zero dose as the studied vaccination outcome, therefore focusing on the most vulnerable children; the national profiles created, with the prevalence of key indicators and national decision trees; and the novel use of decision trees. They are a versatile tool that can be used for identifying risk groups of other outcomes in the context of global and public health. The CART implementation, in particular, can thrive in the large sample sizes that are quite common in the field. Furthermore, it allowed for the inclusion of 17 indicators of vulnerability without previous specification of cut-off points and interactions, dealing with the complex intersections between them. 

Unfortunately, we were able to identify risk groups for only 25 out of the 92 countries studied. The tree’s inability to identify risk groups is most likely due to three factors (individually or in combination): the indicators selected for the analyses failing to discriminate zero-dose children from the remaining children in the sample, low sample size, and low zero-dose prevalence. The last two factors impact the algorithm’s ability to create groups that are large enough and with enough zero-dose children to be maintained during tree creation, especially with our imposed limitation of at least 50 children in each group. This is supported by the fact that countries where risk groups were identified had a median zero-dose prevalence more than six times higher and a median sample size more than 60% higher than the countries with no risk groups identified. Furthermore, countries with really low zero-dose prevalence may not have well-defined risk groups from a population perspective and the remaining children might only be reached by population level interventions. Nevertheless, for the 25 countries where risk groups were identified, the result was similar to the pooled tree analysis: antenatal care visits, delivery in a health facility, and tetanus vaccination were the most common indicators selected by the trees; the last one together with the wealth indicator. 

There are several limitations to these analyses. First, since children with missing information in a vaccine are considered as non-vaccinated, the zero-dose prevalence might be overestimated. As a sensitivity analysis, we calculated the pooled zero-dose prevalence removing all zero-dose children with missing information for any vaccine, resulting in a prevalence of 7.4% (7.1–7.6%). Among the top five countries, South Sudan and Congo DR had 21% and 18% of all zero doses from children with missing information in at least one vaccine. Although this is far from ideal, it follows the current recommendation by the WHO on how to treat missing information for vaccination records [9] and is a more conservative approach, considering the risks of non-vaccination. 

Second, although we were successful in identifying zero-dose risk groups in the pooled analysis, 67% of all zero-dose children in the sample were classified as part of the lowest risk group. This may be due to indicators of vulnerability not included in the analysis or due to the fact that some children do not belong to a specific and well-defined risk group. Further investigation is necessary—possibly using other indicators—as well as a general strengthening of health services, combined with population interventions, in order to ensure that those children are not missed. 

Third, the highest zero-dose prevalence found in a risk group from the pooled tree was 42%. One could argue that this is a low prevalence for a more traditional application of a classification algorithm, especially if used as a prediction tool. Considering that our main goal was to identify the risk groups, not to create predictions, and since zero-dose prevalence tends to be really low in most countries, we believe that 42% represent a significantly high prevalence from a public health perspective.

Fourth, the choice of using a double misclassification cost was arbitrary and the final tree is dependent on that choice. As a sensitivity analysis, we tested other adjustment weights (1, 3, 4, and 5). For 1, no risk groups were identified. For 3 to 5, a very similar tree was created, but the last split (if the child was born in a health facility) was not present. Therefore, the final message remains valid, but with the caveat that the last split should be interpreted with caution. 

## 5. Conclusions

“Leave no one behind” is an opportunity to reach not only those who have been left out by vaccination, but also by other primary healthcare services. The children at higher risk of being zero dose are also the ones whose mothers were left out by antenatal care, delivery, and vaccination services. Those children are also poorer, more rural, and their mothers less educated. Further integration of primary health care services and targeted interventions toward the most vulnerable communities are necessary in order to achieve the Sustainable Development Goals and the Immunization Agenda 2030.

## Figures and Tables

**Figure 1 vaccines-09-00646-f001:**
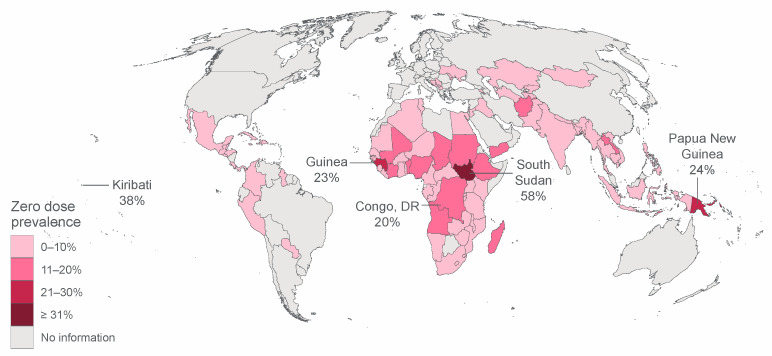
Zero dose national prevalence map and top five countries with highest prevalence.

**Figure 2 vaccines-09-00646-f002:**
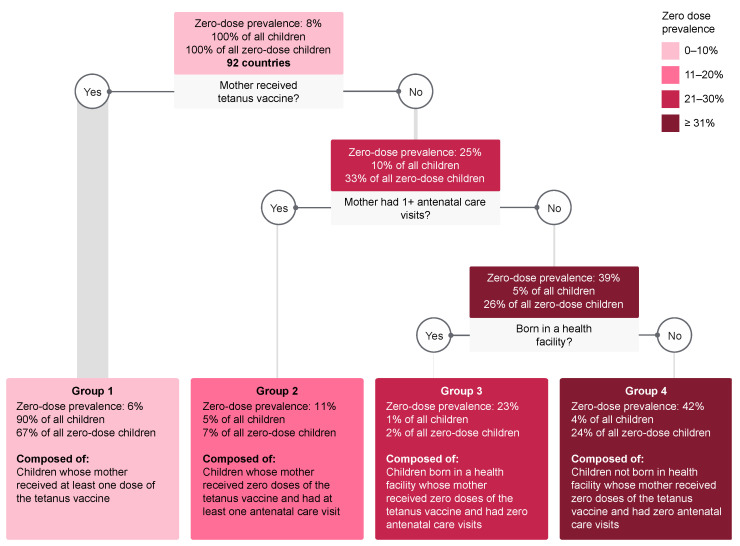
Decision tree for the combined sample of 92 countries.

**Figure 3 vaccines-09-00646-f003:**
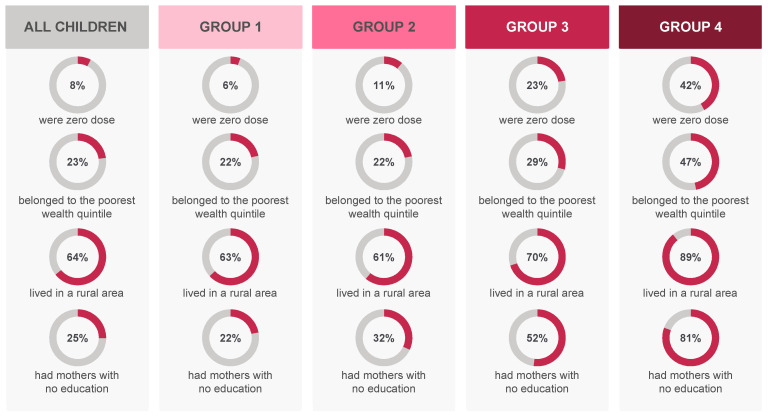
Description of the four zero-dose groups according to key inequality indicators.

**Table 1 vaccines-09-00646-t001:** Indicators featured in national decision trees.

Indicator	Number of National Trees That Featured the Indicator (out of 25 Trees)
Number of antenatal care visits	15
Place of delivery	10
Mother’s tetanus vaccine doses	8
Wealth quintiles	8
Number of household members	5
Mother’s age	4
Mother’s education	4
Religious group	4
Ownership of radio	2
Place of residence	1
Child’s sex	1
Number of live born children that had died	1
Number of living children	1
Ownership of TV	1
Sex of household head	0
Mother’s marital status	0
Ownership of refrigerator	0

## Data Availability

All the analyses were carried out using publicly available datasets that can be obtained directly from the DHS (dhsprogram.com) the MICS (mics.unicef.org) websites. Datasets are continuously sourced and updated by the International Center for Equity in Health (equidade.org) as they are released. We used the last available versions in 7 August 2020.

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
