# Peer review of "Finding Children with High Risk of Non-Vaccination in 92 Low- and Middle-Income Countries: A Decision Tree Approach"

_vaccines, 2021, doi:10.3390/vaccines9060646_

Round 1
Reviewer 1 Report
The authors examined how to identify high-risk of non-vaccination in low- and middle-income countries. This is an important study with potential to add to the literature. The methods were reported in detail to allow for replication. Most importantly, the study used a large, population-based study with national coverage, that is widely perceived to be of high quality, as they were based on sound sampling methodology with high response rate. In addition, the DHS survey questionnaires were similar across countries yielding inter-country comparable data.
My main concern is how good are the models in identifying high-risk for non-vaccination, what are the predictive power of those models? The authors should consider holding out some sample to calculate model accuracy measures such as: the area under the receiver operating characteristic curve (AUC), sensitivity, specificity, positive predictive value, negative predictive value, and accuracy value.
Other minor comments:
- How was the pooled zero-dose prevalence calculated, meta-analysis?
- For each group, three percentages were reported: (1) zero dose prevalence, (2) % of all children and (3) all zero-dose children. The difference between zero dose prevalence and all zero-dose children is not clear?
Reviewer 2 Report
- Please, follow in the abstract a natural order: firstly, a rationale to carry out the study, then, the objective, after that methods, results, discussion and conclusion. I am talking about the contents, because I guess the journal has its editorial format.
- Try to use MESH terms as keywords.
- The methods section is very clear and I would be able to replicate your results, which is the most important point in the reporting of a scientific work.
- Why did the authors combine all the vaccines in one outcome? Maybe, they could do the calculations separately for each type of vaccine, because there are possibilities of differences. This could be interesting to be assessed.
- Why 1800 children in the sample size as a cut-off point?
- The rest of the paper I think it is fine.
Reviewer 3 Report
Estimated Authors,
I've read with great interest the present article entitled "Finding children with high risk of non-vaccination in 92 low- and middle-income countries: a decision tree approach".
The study group lead by Santos has performed an innovative analysis of a very classic topic, and albeit the results of this research are not radically new, their significance is evident.
Pros and cons, as well as relative limitations of this study have been properly addressed in the discussion, whose content is up to date in terms of references and confrontation with available evidences.
In my opinion, the present paper may be accepted as it is.
